# Evaluating Disparities in the Quality of Post hoc Explanations when the Explained Blackboxes are subjected to Fairness Contraints

## Abstract

In recent years, the application of machine learning models in critical domains has raised significant concerns regarding the fairness and interpretability of their predictions. This study investigates the disparities in the quality of post-hoc explanations generated for complex black-box models, specifically focusing on the influence of fairness constraints on these explanations across diverse demographic groups. Utilizing datasets from ACSIncome, AC-SEmployment, and COMPAS, we employ explanation methods such as LIME and KernelSHAP to evaluate metrics including Maximum Fidelity Gap from Average (MFGA), Consistency and Stability. Our findings reveal that the imposition of fairness constraints impacts the fidelity and consistency of explanations, with notable variations observed between demographic groups. While some datasets demonstrate equitable explanation quality across genders, significant biases persist in others, particularly affecting younger individuals and racial minorities. The research highlights the necessity for robust fairness-preserving techniques in post-hoc explanations and underscores the critical need for transparency in AI-driven decision-making processes. By correlating model unfairness with disparities in explanation quality, this work aims to contribute to the ongoing discourse on ethical AI, emphasizing the importance of both accuracy and fairness in machine learning applications. The source code is available at

https://anonymous.4open.science/r/Fairness-via-Explanation-Quality-60F1/

## 1 Introduction

Over the past decades, machine learning methods have become powerful and increasingly applied in high-stakes domains like health, education and justice (Chen et al., 2021; Jamison, 2017; Tuggener et al., 2019; Alarie et al., 2016). This has also led to an increase in the deployment of "black boxes". The latter have been recognized for their intricate and often non-transparent decision-making processes, which present difficulties in grasping the reasoning behind specific decisions. This opacity becomes a critical issue, particularly when such models unintentionally foster biases or unequal treatment of certain groups Doshi-Velez & Kim (2017) and so the need to interpret and explain these models has become a vital issue for practitioners and decision-makers.

Addressing these concerns has led to an increasing emphasis on creating methods for post-hoc explanations. Post hoc explanation methods can be categorized into four main groups: counterfactual Wachter et al. (2017), perturbation-based Ribeiro et al. (2016); Slack et al. (2021); Plumb et al. (2018), gradient-based Selvaraju et al. (2017); Smilkov et al. (2017) and rule-based Ribeiro et al. (2018). Counterfactual explanations involve searching for instances in high-dimensional feature spaces, which can be computationally demanding and may propose changes that are impractical to achieve in real-world scenarios (Laugel et al., 2019). Gradient-based methods, while commonly applied to unstructured data like images, have limitations such as sensitivity to input noise and difficulty in detecting spurious correlations, occasionally producing visually similar explanations for different (Adebayo et al., 2021). Rule-based methods sometimes yield complex and

difficult-to-understand rules, especially with high-dimensional data, and finding the most effective rule can be computationally intensive.

Perturbation-based post hoc explanation methods, notably LIME (Local Interpretable Model-agnostic Explanations), have gained prominence, particularly for tabular data. Another vital point is ensuring the quality of these post-hoc explanations. Many research works focus on this and include metrics such as Mufidelity, Deletion and Insertion scores and Average Stability (Fel et al., 2022; Zhou et al., 2021). Recent studies have highlighted disparities in the fidelity of these methods across different demographic groups, such as 'gender' and 'race' (Allgaier et al., 2023; Dai et al., 2022). To tackle these issues, Dai et al. (2021) proposed a fairness-preserving approach for LIME, incorporating fairness constraints into its objective function, drawing on previous work that enhanced fairness in machine learning through similar constraints. Simultaneously, Balagopalan et al. (2022) introduced a robust LIME explanation model using the 'Just train twice' methodology, demonstrating fidelity improvements in certain cases and datasets, particularly with neural network methods. Such methods are designed to demystify and simplify the decision-making of these intricate black-box models. Offering transparent explanations is crucial to guarantee that users can have confidence in and competently manage AI systems, ensuring that the decisions made are equitable and defensible (Bharati et al., 2023; Rajabi & Kafaie, 2022; Khosravi et al., 2022).

It is worth noting the evidence indicating that factors like sample size, covariate shift, concept shift, omitted variables as well as data and model properties can influence the accuracy of model predictions and contribute to disparities in the performance of black box models, particularly, Mhasawade et al. (2024) demonstrate that an increase in covariate shift, concept shift, and omission of covariates amplify explanation disparities, with the effect pronounced higher for neural network models that are better able to capture the underlying functional form compared to linear models. Notably, imbalances in sample size have been associated with biases in prediction Kleinberg et al. (2022) and calibration models Agustina Ricci Lara et al. (2023). Additionally, insufficient samples from specific subgroups are recognized to impact model performance and the generalizability to those particular subgroups (Cai et al., 2023; Chen et al., 2023). Another source of algorithmic unfairness stems from unevenly missing data across subgroups Wang & Singh (2021); Martínez-Plumed et al. (2019), which can lead to both an imbalance and a sample that doesn't accurately represent the true distribution of the target population, thereby inducing a distribution shift for certain subgroups (Pessach & Shmueli, 2023).

However, it's crucial to note that fairness challenges in machine learning models, such as prediction models, are multifaceted, potentially arising from issues within the data, the black box models themselves, or their interpretation (for instance, through explanation techniques) (Gebru et al., 2021; Barocas et al., 2023).

## 2 Related Works

### 2.1 Explainable Machine Learning

Numerous machine learning and deep learning algorithms have been created, with continuous progress being made in the field. The latter category of algorithms is particularly advanced and yields impressive outcomes. Consequently, there is a rising trend in the development and implementation of black box models in practical applications, which has sparked an increased interest in the development of explanations (usually due to ethics Bhatt et al. (2020b); Holzinger (2018); Roscher et al. (2020) or laws Bibal et al. (2021) ) that summarize the behaviours of these black boxes.

Model-agnostic explainability techniques can be broadly categorized into local and global methods (Du et al., 2019). Local approaches focus on explaining individual predictions made by a model. They do this by approximating the model's decision-making process near a specific data point (Plumb et al., 2018; Ribeiro et al., 2018; Botari et al., 2020; Rathi, 2019). The weights derived from these local models then elucidate the reasoning behind the predictions of the more complex model.

LIME (Local Interpretable Model-Agostic Explanation), is an explanation technique that explains the predictions of any classifier in an interpretable and faithful manner, by learning an interpretable model locally around the prediction (Ribeiro et al., 2016). KernelSHAP, a modification of SHAP(Shapley Additive ex-

Planations) Lundberg & Lee (2017) is an explanation algorithm that assigns each feature an importance value for a particular prediction by making use of some weighting Kernel function. The above-mentioned algorithms are examples of attribution-based explanations.

## 2.2   Algorithmic Fairness and Bias Mitigation

Formalizing fairness is rapidly expanding in research. Current research primarily focuses on defining fairness on either an individual or group basis  (Berk, 2017; Chen et al., 2018; Chouldechova, 2017; Chouldechova & Roth, 2018; Zafar et al., 2017; Zemel et al., 2013).

Individual fairness, as described in Dwork et al. (2012), entails making consistent predictions for similar individuals. In contrast, group fairness involves ensuring equitable predictions across different demographic groups, such as gender, age or race.

Our study is centered on group-level fairness in binary classification, specifically measured by the demographic parity difference (DPD) also called statistical parity difference and equalized odds difference, commonly used fairness metrics in group fairness. We employ a probabilistic approach to define this metric, which allows for comparing gaps between groups. DPD is given mathematically by

$$\text{DPD} = \max_a \left| \mathbb{E}\left[h(X)|A = a\right] - \mathbb{E}[h(X)] \right| \tag{1}$$

and EOD by

$$\text{EOD} = \max_{a,y} |\mathbb{E}\left[h(X)|A = a, Y = y\right] - \mathbb{E}[h(X) \mid Y = y]| \tag{2}$$

where $h(X)$ represents a predictor, $a$ is a sensitive attribute and $Y$ the ground truth label. To enhance group fairness, three main strategies are typically employed Caton & Haas (2020); modifying data beforehand to reduce bias (preprocessing) Paul & Burlina (2021), incorporating fairness during model training (in-processing/reduction) and post-processing approach conducted after training by utilizing a holdout set that was excluded from the model's training process  (Mehrabi et al., 2021).

We apply a reduction approach method suggested by Agarwal et al. (2018) to train fair blackbox models. Moreover, recent studies have indicated that training models robust to group differences can enhance fairness by improving accuracy for the least advantaged groups. The fairness constraint is grounded on a utility metric, $\varphi$, which can be assessed for individual data points and is averaged across different data point groups to establish the utility parity constraint. The utility implements constraints that permit some degree of violation of the utility parity constraints, with the maximum allowed violation specified as a difference. The relaxation of this utility parity constraint based on differences can be represented by

$$\varphi_{a,e} - \varphi_e \leq 0 \text{ or } \varphi_e - \varphi_{a,e} \leq 0$$

and subsequently replaces zero on the right-hand side with a value designated as **difference bound**  (Agarwal et al., 2018). The value of difference bound restricts the difference between the utility of each group and the overall mean utility within each event.

## 2.3   Group-Based disparities Post hoc explanation quality

In their work, Dai et al. (2022) began exploring group-based disparities in explanation quality, identifying several key properties such as fidelity (accuracy), stability, consistency, and sparsity that contribute to explanation quality. They discussed why disparities in these properties are problematic and proposed an evaluation framework to quantitatively measure them. Through empirical analysis across three datasets, using six post-hoc explanation methods and various model classes, they found that group-based disparities in explanation quality are more likely to occur when models are complex and non-linear. Notably, explanation methods such as Integrated Gradients and SHAP were more prone to exhibit disparities, highlighting previously unexplored ways in which explanation methods can introduce unfairness in decision-making processes.

Similarly, Balagopalan et al. (2022) evaluated explanation model fairness using the fidelity gap and demonstrated that improving explanation fairness can significantly enhance decision-making for underserved groups.

They observed notable differences in the fidelity of explanation models among subgroups across two black-box model architectures and four popular explanation methods.

Further addressing these challenges, Mhasawade et al. (2024) examined how data properties (e.g., limited sample size, covariate shift, concept shift, and omitted variable bias) and model characteristics (e.g., sensitive attribute incorporation and functional form selection) contribute to explanation disparities. Through simulated scenarios and experiments, they found that heightened covariate and concept shifts, as well as the omission of covariates, amplify explanation disparities, particularly in neural network models. Their findings suggest that disparities arise from both data and model characteristics.

Building on this foundation, the motivation for this work arises from the need to better understand how these explanation disparities manifest across demographic groups, especially when post-hoc methods are applied to complex black-box models. While previous research has highlighted the influence of factors such as model characteristics, data properties, and fairness on explanation quality, it has typically been limited to a single level of fairness. This research takes a more granular approach by investigating 99 different levels of unfairness. Experimenting with multiple fairness levels, we seek to capture the broader relationship between fairness constraints and explanation disparities. The study also aims to determine if there are systematic correlations between unfairness and disparities in explanation quality. By analyzing the relationship between varying levels of unfairness and the quality of explanations across different demographic groups, the research seeks to uncover whether these disparities are systematically linked to the degree of unfairness in the model. The findings will provide deeper insights into how fairness constraints may influence explanation quality and whether addressing unfairness can enhance transparency and equity in AI-driven decision-making processes. We do this by answering the following questions:

- **RQ1** How does the unfairness of black-box models affect the quality of post hoc explanations in different demographic groups?

- **RQ2** To what extent is there a significant correlation between model unfairness and disparities in explanation quality metrics across different groups?

- **RQ3** How do different fairness metrics impact the relationship between model unfairness and explanation disparity?

## 3 Measuring Fairness of explanations

### 3.1 Notation

Consider a dataset $\mathcal{D} = \{(x_1, y_1), (x_2, y_2), \cdots (x_n, y_n)\}$ composed of **n** training data points where $x_i$ is a $d$-dimensional feature vector for the $i^{th}$ data point in $\mathcal{D}$ and $y_i \in \mathcal{Y}$ is the corresponding binary class label. Each data point comprises some sensitive feature/attribute **a**. Consider also, a complex and non linear blackbox classifier $h : \mathbb{R}^d \to \{0, 1\}$, and a global explanation method $E : (x, h) \to \psi \in \mathbb{R}^d$, where $\psi$ represents the output vector of feature importance. We train an explanation model $E$ chosen from the set of interpretable models like linear models or decision trees to approximate the behaviour of $h$ in the vicinity of $x_i$.

### 3.2 Explanation Quality

Although research in explainable machine learning is expanding, there's yet to be an agreed-upon method for gauging a model's explainability quality (Poursabzi-Sangdeh et al., 2021). Consequently, human assessment remains the primary method for evaluating the quality of a model's explanation, focusing on factors like transparency, the trust it engenders in users, or how well humans understand the model's decisions (Petsiuk et al., 2018). Here we use one fidelity metric described by Balagopalan et al. (2022), a consistency and stability metric (Dai et al., 2022). Fidelity measures how accurately an explanation model replicates the predictions of a black box model and facilitates the assessment of fairness by demonstrating the alignment between the machine learning model and the explanation model. Mathematically, it is given by $\frac{1}{N} \sum_{i=1}^{N} \mathcal{P}(h(x_i), E(x_i))$, where $\mathcal{P}$ is a performance metric such as mean squared error. Here, we categorize the features into continuous and categorical types. We calculate the median values for the continuous features, and for the categorical

features, it identifies the least frequent values. This information is crucial for modifying the least important features during the explanation process.

We select the $top_k$ features based on their attribution scores, sorting the feature attributions and determining the indices of these top features. The least important features are identified by subtracting the top features from the full set of features. Subsequently, the identified least important features are modified: Gaussian noise is added to the median value for continuous features to introduce variability, while for categorical features, the values are replaced with the least frequent category. The result is a modified dataset that retains the most significant features while altering only the least important ones based on their attributions.

### 3.2.1 Maximum Fidelity Gap from Average (MFGA)

The Maximum Fidelity Gap from the Average quantifies the greatest deviation in fidelity for any group compared to the average fidelity across all groups. This metric evaluates how much the fidelity of an explanation model for disadvantaged groups differs from the overall average fidelity. The maximum fidelity gap from the average, $\Delta Q_m$, is represented as follows:

$$\Delta Q_m = \max_j \left[ \frac{1}{N} \sum_{i=1}^{N} \mathcal{P}(h(x_i), E(x_i)) - \frac{1}{N_j} \sum_{i:\delta_i^j=1} \mathcal{P}(h(x_i), E(x_i)) \right] \qquad (3)$$

where $N$ denotes the total number of data points, $\delta_i^j = 1$ indicates that point $x_i$ belongs to the $j$-th group, and $N_j$ represents the number of data points where $\delta_i^j = 1$.

Let's denote the individual fidelity gap by $\Delta Q^j$ such that $\Delta Q_m = \max_j \Delta Q^j$. Our focus is on individual and maximum fidelity gap from the average for the accuracy performance metric, following (Mhasawade et al., 2024). This involves assessing the accuracy between the predictions of the black box model $h(\cdot)$ and the explanation method $E(\cdot)$.

## 3.3 Relative Consistency

Consistency embodies the idea that when an explanation for a single data point is computed multiple times, each iteration should yield similar results. Discrepancies among explanations for the same input $x$ hint at potential unreliability in these explanations (Fel et al., 2022). If a single point can generate a plethora of highly divergent explanations, it's improbable that any one explanation is accurate, signifying poor approximation quality. To quantify consistency for a given point $x$, we generate several explanations for that point and then compute the average euclidean distance between the initial explanation and each subsequent one. Let $E$ be an explanation method, $d$ is the euclidean distance, we define the inconsistency by

$$\text{Inconsistency Score} = \frac{2}{n_{\text{expl}}(n_{\text{expl}} - 1)} \sum_{i=1}^{n_{\text{expl}}} \sum_{j=i+1}^{n_{\text{expl}}} d(E_i, E_j) \qquad (4)$$

### 3.3.1 Stability

The principle that similar points should receive similar explanations often termed stability, robustness, or insensitivity has been extensively discussed (Dai et al., 2022). If different explanations are provided for similar points, it suggests that not all of those explanations can be correct. To assess the stability of an explanation at a point $x$, for each input, noise is randomly introduced and explanations are generated for these perturbed inputs. The metric is the mean $L1$ distance between the explanations of the original and the noise-altered inputs (Bhatt et al., 2020a).

$$\text{Instability}(x, h, E) = \mathbb{E}\left[\|E(x, h) - E(\tilde{x}, h)\|_1\right].$$

We approximate instability empirically by generating $\tilde{x}_j$ $m$ times, producing $\{\tilde{x}_j \mid 1 \leq j \leq m\}$

$$\text{Instability}(x, h, E) \approx \frac{1}{m} \sum_{j=1}^{m} \|E(x, h) - E(\tilde{x}_j, h)\|_1. \qquad (5)$$

# 4   Impacts of quality metrics in real life and how it affects decisions

## 4.1   Maximum Fidelity Gap from Average

The Maximum Fidelity Gap from the Average quantifies the largest disparity in fidelity between a specific group and the overall average fidelity across all groups. This metric is crucial for identifying how much worse the explanations are for disadvantaged groups compared to the average explanation quality. It helps in detecting potential biases where certain groups receive explanations that are significantly less accurate or less reliable, which could lead to unfair decision-making. Addressing the maximum fidelity gap ensures that all demographic groups receive equitable explanations, thereby supporting fairness in model interpretability and promoting trust among all users, regardless of their group affiliation.

## 4.2   Relative Consistency

Relative Consistency measures the reliability of explanations by ensuring that multiple iterations of explanations for the same input do not diverge significantly. This reliability is crucial for reducing the likelihood of contradictory explanations, which can undermine the trust and reliability of the model. By ensuring that the model's explanations are consistent, this metric helps build user trust and makes the model's decisions more understandable and predictable. Furthermore, maintaining high standards in explanation quality is essential in critical applications where the consequences of decisions are significant, such as in healthcare or finance.

## 4.3   Stability

Stability, also known as robustness or insensitivity, evaluates the consistency of model explanations when small perturbations are introduced to the input data. A stable model provides similar explanations for similar inputs, which is vital for ensuring that the explanations are reliable and trustworthy. Instability, where similar inputs lead to widely varying explanations, can signal that the model's explanations are not dependable, potentially leading to confusion or mistrust among users. Stability is particularly important in applications where decisions based on the model's explanations have significant consequences, such as in legal or medical contexts. By ensuring high stability, models can provide consistent and reliable insights, thereby enhancing user confidence in the model's decisions. decisions.

# 5   Analysis

## 5.1   Experimental Setup

### 5.1.1   Data

For the experiment, we chose three benchmark datasets to ensure diversity in the domains they represent, particularly ACSIncome (which predicts if an individual has an income level greater than 50000), ACSEmployment that predicts if an individual is employed or not Ding et al. (2021) and COMPAS (assess the likelihood of recidivism) (Barenstein, 2019). These datasets were selected due to their relevance to societal and economic factors and their availability for research purposes. Using **gender**, as protected attribute for ACSIncome, **age** for ACSEmployment and **race** for COMPAS, we evaluate disparities in explanation quality. To ensure the robustness and generalizability of our findings, the datasets are divided randomly into training, testing, and explanation sets in a ratio of 4:3:3. This stratified division ensures that each subset maintains the same distribution of target labels, thereby preventing any imbalance that could affect the performance evaluation.

### 5.1.2   Models

We consider 2 complex models to evaluate whether explanation disparities occur. Specifically, Xgboost (XGB) and Random Forest Classifier (RF) and carried out cross-validation with GridSearchCV with a max-depth in the range of 1 to 7.

Table 1: Datasets and Sensitive Attribute used in our experiments

| Dataset | Outcome Variable | Sensitive attribute |
| --- | --- | --- |
| ACSIncome | Income > 50 K | Gender (2 groups) |
| ACSEmployment | Employment status recode | Age (2 groups) |
| COMPAS | Defendant re-offends | Race (2 groups) |

### 5.1.3 Bias Mitigation

For bias mitigation, we adopted a reduction approach known as Exponentiated Gradient, Agarwal et al. (2018); Bird et al. (2020) which converts a binary classification task into a problem cost-sensitive classification. This transformation allows for the selection of a randomized classifier that achieves a minimal error rate while adhering to fairness constraints (Mbiazi et al., 2023). We mitigate the model for fairness using demographic parity as a constraint. To assess the impact of fairness constraints on explanation quality, we vary the difference bound in a range from 0.01 to 1 with a step size of 0.01 to obtain 99 fairer versions of the base model. By varying the degree of fairness enforcement, we can evaluate how different levels of fairness affect the explanation quality provided by our models.

### 5.1.4 Hyperparameters

We run Lime and KernelShap with 100 perturbations generated around each input data point and batch size of 64.

### 5.1.5 Hypotheses

We perform the Mann-Whitney U test to compare the distributions of the quality metrics between different groups. The test is used to determine whether the distributions of two independent groups are different. The p-values shown indicate whether the differences in these distributions are statistically significant at a 95% confidence interval. We carry out this for fidelity, relative consistency and stability. This is done with the following hypotheses;

- **Null Hypothesis ($H_0$):** There is no significant difference in the distribution between the groups for the given dataset, model, and explanation method.

- **Alternate Hypothesis ($H_A$):** There is a significant difference in the distribution between the groups for the given dataset, model, and explanation method.

For the Correlation Test, we are conducting a Spearman's correlation test to assess the relationship between model unfairness and the disparity in quality metrics, this is done with the following hypotheses;

- **Null Hypothesis ($H_0$):** There is no significant monotonic relationship between model unfairness and disparity in quality metrics

- **Alternate Hypothesis ($H_A$):** There is a significant monotonic relationship between model unfairness and disparity in quality metrics.

The Spearman's correlation test is carried out at a 95% confidence level, meaning if the p-value is less than 0.05, the null hypothesis will be rejected, suggesting that there is a statistically significant monotonic relationship between model unfairness and the disparity in quality metrics. We carry out this for the four explanation metrics.

## 5.2 Setting and Implementation

We start by splitting each dataset into a training set, test set and explanation set with respect to some random seed as mentioned above in 5.1.1. To enhance model fairness, we train $h$ using a reduction approach,

exponentiated gradient to mitigate bias as mentioned in 5.1.3 above. The fairness of models is assessed using the test dataset. Subsequently, the explanation dataset was split into two subsets based on the sensitive attribute (with values 0 and 1 for males and females respectively), age attribute ( with 0 representing individuals with less than or equal to 30 and 1 otherwise) and race attribute (0 for African-American and 1 for Caucasian). For the explanation method, and for each mitigated version of the base model, we compute the fidelity and average stability score. To assess if the metrics differ significantly, we carry out 7 trials, repeating the computations for different random seeds. For each random seed, explanation method and percentage violation in fairness metric we record the corresponding median metric value.

## 5.3 Results

After applying fairness constraints to our base models, how sure are we that the bias mitigation is effective? Figures  7a and  7b show how the demographic parity difference changes with respect to percentage in fairness violation for each of some models and dataset combination. The variation is seen to be increasing because in adding the percentage violation, we move from a less to a more unfair model.

**RQ1**

**Maximum Fidelity Gap from the Average:** Figures  1 and  8 provided, show the plots of the individual deviation (for each subgroup) from the overall fidelity ($\Delta Q^j$). For **ACSIncome** dataset, both XGB and RF models show relatively small fidelity gaps between males and females. The CDF curves for these two groups are closely aligned, indicating similar explanation fidelity across genders. The p-values for the Mann-Whitney U test confirm this observation, with values as high as **0.29** for XGB and **0.1** for RF with Lime, and **0.29** for XGB and **0.1** for RF in KernelShap as can be seen in tables  2 and  3. These higher p-values suggest no statistically significant difference in fidelity between demographic groups, indicating that the explanations provided by both models are fair across genders. In contrast, for the ACSEmployment and COMPAS datasets, there are significant differences in explanation fidelity across demographic groups. For the **ACSEmployment** dataset, younger individuals (age $\leq 30$) experience larger fidelity gaps compared to older individuals, as evidenced by the low p-values, such as $1.97 \times 10^{-7}$ for XGB and $4. \times 10^{-8}$ for RF in Lime, and similarly low values for KernelShap. The COMPAS dataset shows significant racial disparities in fidelity, with the Caucasian group experiencing consistently higher gaps. The p-values, like 0.042 for XGB and 0.0002 for RF in LIME, confirm that explanation fidelity differs significantly between racial groups, suggesting that post-hoc explainability methods provide less accurate explanations for certain groups.

**Inconsistency score:** Figures 2 and 9 show cumulative distribution functions (CDFs), and the Mann-Whitney U test p-values from Tables 4 and 5 provide insights into the explanation inconsistency of XGBoost (XGB) and Random Forest (RF) models across demographic subgroups. For the **ACSIncome** dataset, both XGB and RF models yield extremely low p-values (on the order of $10^{-34}$) for both LIME and KernelSHAP, indicating a highly significant difference in explanation consistency between demographic groups. This suggests that one group consistently receives more reliable explanations than the other, as confirmed by both the CDF plots and statistical tests. In the **ACSEmployment** dataset, the p-values for both XGB and RF models are also very low (e.g., $9.27 \times 10^{-3}$ to $2.8 \times 10^{-4}$ for DPD, and as low as $5.33 \times 10^{-5}$ for EOD), again indicating statistically significant differences in explanation consistency between age groups. The gap is more pronounced for XGB than RF, as reflected in the lower p-values for XGB. For the **COMPAS** dataset, XGB shows low p-values for both LIME and KernelSHAP (e.g., $1.69 \times 10^{-3}$ and $1.65 \times 10^{-3}$ for DPD, $1.51 \times 10^{-4}$ and $2.03 \times 10^{-4}$ for EOD), indicating significant differences in explanation consistency between racial groups. In contrast, RF yields higher p-values (e.g., 0.122 and 0.229 for DPD, 0.0921 and 0.179 for EOD), suggesting no statistically significant difference in explanation consistency between groups for this model.

In summary, the p-values confirm that significant disparities in explanation consistency exist across demographic subgroups for most datasets and models, with the ACSIncome and ACSEmployment datasets showing the most pronounced and consistent gaps, particularly for XGB. For COMPAS, the gap is significant for XGB but not for RF.

**Instability score:** The analysis of explanation instability across demographic subgroups, based on the CDFs from 3 and  9 and Mann-Whitney U test p-values (tables  6 and  7), reveals significant differences in the stability of explanations provided by XGBoost (XGB) and Random Forest (RF) models in the ACSIncome,

ACSEmployment, and COMPAS datasets. For the **ACSIncome** dataset, both XGB and RF models yield extremely low p-values (on the order of $10^{-34}$) for both LIME and KernelShap, indicating highly significant differences in instability scores between demographic groups. The CDF curves suggest that one group (e.g., females) consistently receives more stable explanations than the other (e.g., males), with the gap being present in both models. In the **ACSEmployment** dataset, the p-values for XGB are 0.0694 (LIME) and 0.0405 (KernelShap) for DPD, and 0.00518 (LIME) and 0.01414 (KernelShap) for EOD. For RF, the p-values are 0.0469 (LIME) and 0.0543 (KernelShap) for DPD, and 0.000749 (LIME) and 0.000697 (KernelShap) for EOD. These results indicate that explanation instability differences between age groups are statistically significant for most cases, especially for EOD, with the gap being more pronounced in XGB than RF. For the **COMPAS** dataset, both XGB and RF models show extremely low p-values (e.g., $5.23 \times 10^{-17}$ and $5.45 \times 10^{-34}$ for LIME in DPD, $2.69 \times 10^{-14}$ and $8.33 \times 10^{-34}$ for KernelShap in DPD, and similarly low values for EOD), confirming significant instability differences between racial groups. The results suggest that both models provide less stable explanations for certain groups, with RF showing particularly pronounced instability.

Overall, the p-values confirm that significant disparities in explanation stability exist across demographic subgroups for all datasets and models, with the most pronounced and consistent gaps observed in ACSIncome and COMPAS, and with XGB generally showing larger gaps than RF in ACSEmployment.

Table 2: P-values for Mann-Whitney U test for $\Delta Q^j$ at 95% confidence interval for DPD

| Dataset | Model | **Lime** | **KernelShap** |
|---|---|---|---|
| **ACSIncome** | XGB | 0.292 | 0.292 |
| | RF | 0.104 | 0.104 |
| **COMPAS** | XGB | **0.042** | **0.042** |
| | RF | **0.000242** | **0.000242** |
| **ACSEmployment** | XGB | **1.97e-07** | **1.97e-07** |
| | RF | **4.04e-08** | **4.04e-08** |

Table 3: P-values for Mann-Whitney U test for $\Delta Q^j$ at 95% confidence interval for EOD

| Dataset | Model | **Lime** | **KernelShap** |
|---|---|---|---|
| **ACSIncome** | XGB | 0.104 | 0.104 |
| | RF | 0.105 | 0.105 |
| **COMPAS** | XGB | **0.00298** | **0.00298** |
| | RF | **1.15e-06** | **1.15e-06** |
| **ACSEmployment** | XGB | **1e-06** | **1e-06** |
| | RF | **0.000102** | **0.000102** |

Table 4: P-values for Mann-Whitney U test for relative consistency score at 95% confidence interval for DPD

| Dataset | Model | **Lime** | **KernelShap** |
|---|---|---|---|
| **ACSIncome** | XGB | **5.45e-34** | **5.45e-34** |
| | RF | **5.45e-34** | **5.45e-34** |
| **COMPAS** | XGB | **0.00169** | **0.00165** |
| | RF | 0.122 | 0.229 |
| **ACSEmployment** | XGB | **0.00927** | **0.00681** |
| | RF | **0.000282** | **0.000217** |

Table 5: P-values for Mann-Whitney U test for relative consistency score at 95% confidence interval for EOD

| Dataset | Model | **Lime** | **KernelShap** |
|---|---|---|---|
| **ACSIncome** | XGB | **5.45e-34** | **5.45e-34** |
| | RF | **5.45e-34** | **5.45e-34** |
| **COMPAS** | XGB | **0.000151** | **0.000203** |
| | RF | 0.0921 | 0.179 |
| **ACSEmployment** | XGB | **0.00125** | **0.00149** |
| | RF | **5.33e-05** | **5.39e-05** |

Table 6: P-values for Mann-Whitney U test for stability score at 95% confidence interval for DPD

| Dataset | Model | **Lime** | **KernelShap** |
|---|---|---|---|
| **ACSIncome** | XGB | **5.45e-34** | **5.45e-34** |
| | RF | **5.45e-34** | **5.45e-34** |
| **COMPAS** | XGB | **5.23e-17** | **2.69e-14** |
| | RF | **5.45e-34** | **8.33e-34** |
| **ACSEmployment** | XGB | 0.0694 | **0.0405** |
| | RF | **0.0469** | 0.0543 |

Table 7: P-values for Mann-Whitney U test for stability score at 95% confidence interval for EOD

| Dataset | Model | **Lime** | **KernelShap** |
|---|---|---|---|
| **ACSIncome** | XGB | **5.45e-34** | **5.45e-34** |
| | RF | **5.45e-34** | **5.45e-34** |
| **COMPAS** | XGB | **5.81e-20** | **2.88e-18** |
| | RF | **5.45e-34** | **5.45e-34** |
| **ACSEmployment** | XGB | **0.00518** | **0.00414** |
| | RF | **0.000749** | **0.000697** |

**RQ:2**

The correlation between unfairness and disparity in quality metrics ($\Delta Q$) varies across both XGBoost (XGB) and Random Forest (RF) models, displaying non-linear and dataset-dependent patterns. For XGB, there is a clear positive correlation in some cases, such as fig 4a, where higher unfairness corresponds to larger performance disparities across subgroups, indicating that unfair models tend to exhibit more inconsistent quality metrics. However, in other cases, such as fig 4b, a quadratic relationship emerges, with disparity decreasing at first before rising as unfairness increases, suggesting a more complex interaction. Conversely, fig 4c shows an inverse correlation, where higher unfairness surprisingly corresponds to lower performance disparities, potentially reflecting specific subgroup interactions unique to the dataset. For RF, the results show similar variability. In fig 11b, like XGB, there is a positive correlation, where greater unfairness corresponds to larger disparities in quality metrics. However, in fig 11a, a weak inverse correlation suggests that increasing unfairness slightly reduces disparity, while fig 11c shows no significant correlation, implying that unfairness and performance disparities may be independent in this case.

The correlation between disparity in inconsistency scores and unfairness shows different patterns depending on the dataset and model. For XGB, there are strong non-linear relationships, particularly in plot (a) fig 5a and fig 5b, where increasing unfairness first reduces and then increases the disparity in instability. This could indicate a complex interaction between fairness and the reliability of model explanations, with higher unfairness leading to more unstable explanations beyond certain thresholds. In contrast, the RF results show much weaker correlations, with most plots (especially fig 12b) indicating that unfairness has little to no impact on explanation inconsistency. The slight trends observed in some RF plots suggest only a loose connection between fairness and stability in explanations.

Across both models, the relationship between disparity in instability scores and unfairness is non-linear for XGB and weaker for RF. For XGB, several plots (especially fig 6b)) reveal quadratic relationships, suggesting that while moderate unfairness can reduce instability disparity, excessive unfairness might lead to more unstable explanations disparities. This complex interaction indicates that unfairness can both positively and negatively impact explanation stability depending on its magnitude. Conversely, the RF results show weaker linear trends, where unfairness only minimally affects explanation instability. In most cases, increasing unfairness has little to no impact on RF model disparity in stability, suggesting that explanation reliability in RF is more robust to fairness variations compared to XGB.

Table 8: Summary of Correlations in the Figures

| Dataset | Model | **Perfect** | **Slight** | **No Correlation** |
|---|---|---|---|---|
| **ACSIncome** | XGB | 2 | 1 | 1 |
| | RF | 0 | 4 | 0 |
| **ACSEmployment** | XGB | 0 | 4 | 0 |
| | RF | 0 | 3 | 1 |
| **COMPAS** | XGB | 0 | 3 | 1 |
| | RF | 0 | 0 | 4 |

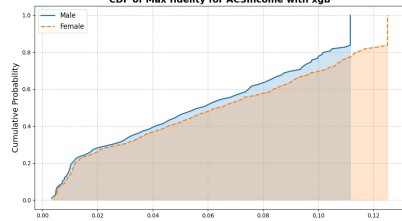 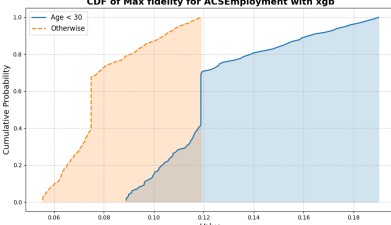 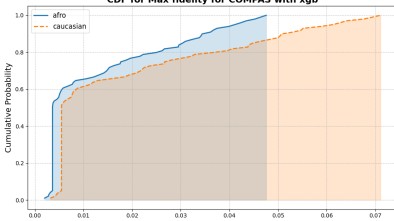

Figure 1: **CDF $\Delta Q^j$ with XGB**

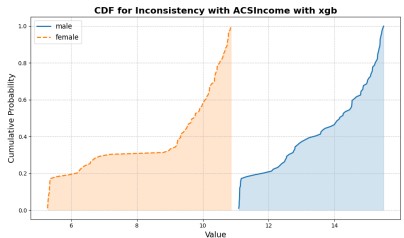 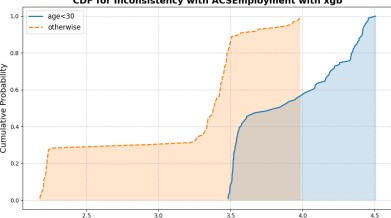 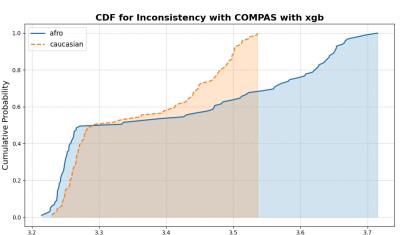

Figure 2: **CDF Inconsistency score with XGB**

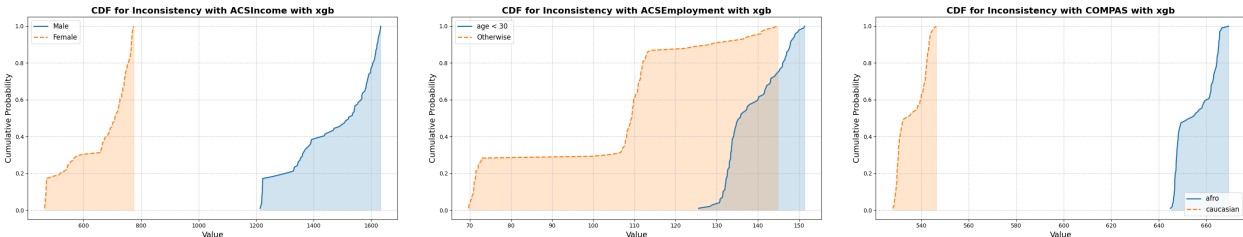

Figure 3: **CDF Instability score with XGB**

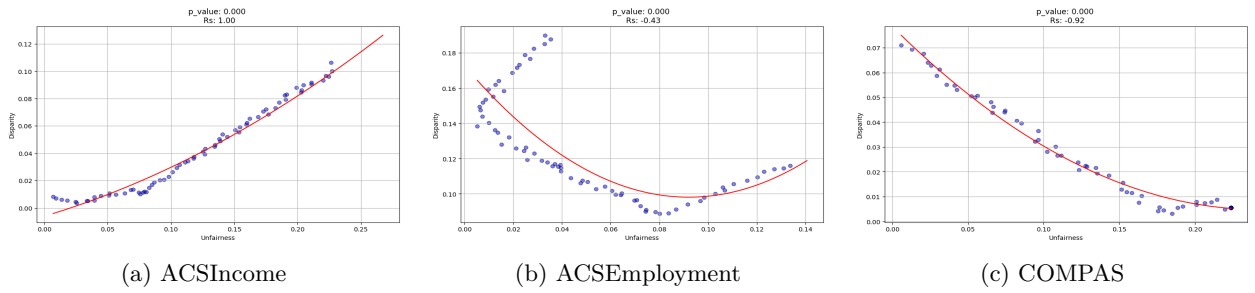

(a) ACSIncome           (b) ACSEmployment           (c) COMPAS

Figure 4: **Correlation for $\Delta Q^j$ and unfairness with XGB**

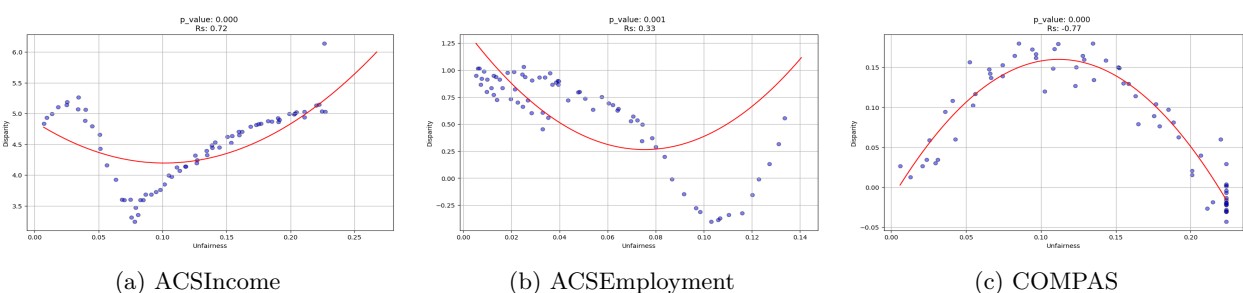

(a) ACSIncome           (b) ACSEmployment           (c) COMPAS

Figure 5: **Correlation for Inconsistency scores and unfairness with XGB**

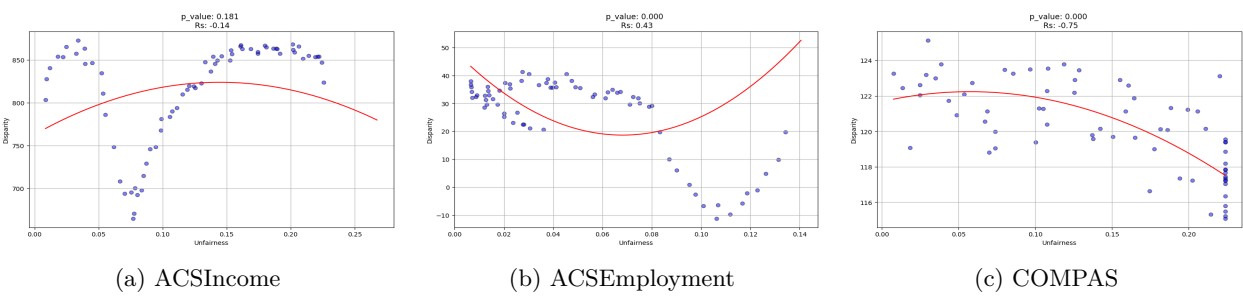

(a) ACSIncome           (b) ACSEmployment           (c) COMPAS

Figure 6: **Correlation for Instability scores and unfairness with XGB**

**RQ:3** Investigation reveals that the choice of fairness metrics Demographic Parity Difference (DPD) and Equalized Odds Difference (EOD) significantly impacts explanation quality disparities, with effects varying across datasets and demographic groups. For the ACSIncome dataset (gender-sensitive), both metrics showed minimal fidelity gaps (p-values $0.104-0.292$), suggesting balanced demographic representation reduces metric

sensitivity. In contrast, ACSEmployment (age-sensitive) and COMPAS (race-sensitive) exhibited substantial disparities, with DPD and EOD producing distinct patterns in explanation gaps (p-values $10^{-6} - 10^{-8}$). The study found that XGBoost models were more sensitive to fairness metric choice than Random Forest models, likely due to their non-linear structure amplifying disparities. Both LIME and KernelSHAP showed similar trends, but their interaction with fairness constraints influenced explanation stability. The analysis uncovered non-linear relationships: DPD constraints often correlated positively with explanation disparity, while EOD constraints sometimes resulted in inverse or quadratic patterns, highlighting how fairness formulations (overall parity vs. conditional parity given labels) fundamentally alter explanation behavior. These findings emphasize that no single fairness metric universally ensures equitable explanations effectiveness depends on dataset characteristics, demographic distributions, and model complexity. Practitioners must prioritize context-aware metric selection, particularly in high-stakes domains like employment or criminal justice, to balance model performance and explanation equity.

## 6 Discussion

The primary objective of the study was to investigate the relationship between the fairness constraints applied to machine learning models and the quality of post-hoc explanations across different demographic groups. The research focuses on the disparities in explanation quality when fairness is enforced on two commonly used black-box models XGB and RF across three datasets: ACSIncome, ACSEmployment, and COMPAS. Explanation quality was assessed using metrics such as Maximum Fidelity Gap from the Average (MFGA), consistency, and stability.

The results showed that disparities in explanation quality are influenced by both the dataset and the model used. For example, in the ACSIncome dataset, the fidelity of explanations was relatively equal across gender groups for both XGB and RF models, as indicated by high p-values from the Mann-Whitney U test, suggesting no significant difference in the quality of explanations between males and females. However, in the ACSEmployment and COMPAS datasets, significant disparities were observed. Younger individuals in the ACSEmployment dataset and non-white individuals in the COMPAS dataset received less accurate and less consistent explanations compared to their counterparts. These findings suggest that while certain models and datasets may provide equitable explanations, others may exhibit significant biases, leading to unfair treatment of certain demographic groups.

The study also revealed that model complexity plays a crucial role in determining the degree of explanation disparity. More complex models like XGB tend to show more pronounced non-linear relationships between unfairness and explanation disparity metrics. For instance, in several cases, increasing unfairness led to larger disparities in explanation consistency and stability, particularly for the XGB model, while RF exhibited weaker correlations in many cases. This suggests that models like XGB, which are more adept at capturing non-linear relationships, may also be more prone to explanation disparities when fairness constraints are not adequately enforced. The choice of fairness metric also played a significant role. Demographic Parity Difference (DPD) and Equalized Odds Difference (EOD) produced different patterns of explanation disparity, with effects varying across datasets and demographic groups. Notably, no single fairness metric ensured equitable explanation quality across all settings, underscoring the need for context-aware metric selection especially in high-stakes domains where explanation equity is critical for trust and accountability.

### 6.1 Threads to valitidy

Several threats to validity may affect the generalizability and robustness of the study's findings. The study focused primarily on two perturbation-based explanation methods, Lime and KernelSHAP, both of which have inherent limitations. Lime relies on a perturbation process that may introduce noise and uncertainty into the explanations, while KernelSHAP's performance can be sensitive to the selection of hyperparameters. Future research should explore other explanation methods, such as gradient-based techniques, to validate the findings across a broader range of explanation methods. Additionally, the study's conclusions are based on three specific datasets ACSIncome, ACSEmployment, and COMPAS each representing different domains and demographic groups. Although these datasets were chosen for their societal relevance, the results may

not generalize to other datasets or domains. The observed disparities in explanation quality could be specific to these datasets, and further research is needed to confirm these findings in other contexts.

## 6.2 Conclusion

The study contributes the growing body of research on fairness and explainability in machine learning by highlighting how disparities in post-hoc explanations can arise even when fairness constraints are applied to the underlying models. The findings emphasize the importance of evaluating not only the accuracy of models but also the fairness of their explanations, particularly in high-stakes domains such as criminal justice and employment, where unfair explanations can have serious real-world consequences. While fairness constraints can reduce disparities in explanation quality for some models and datasets, they are not a one-size-fits-all solution. The extent to which these constraints mitigate explanation disparities varies based on the dataset, model, and specific explanation method used. For example, while fairness constraints improved explanation quality for certain groups in the ACSIncome dataset, they were less effective for younger individuals in the ACSEmployment dataset and non-white individuals in the COMPAS dataset.

These findings suggest that future research and practice should focus on developing more robust fairness-preserving techniques for post-hoc explanations, particularly for complex models like XGBoost, which exhibit more pronounced explanation disparities. Additionally, model developers need to carefully consider the demographic characteristics of the data and the potential for disparities in explanation quality when deploying machine learning models in real-world applications. These results highlight the need to consider fairness not only in model accuracy but also in the quality of explanations provided by machine learning models. Practitioners should integrate a variety of explanation methods, including gradient-based and perturbation-based techniques, to ensure robustness and transparency in different contexts. To address these concerns, organizations deploying machine learning models should adopt broader methods for generating explanations. Relying on a single explanation method can be limiting, and combining Lime, KernelSHAP, and gradient-based techniques will provide more comprehensive and equitable model transparency across diverse demographic groups. Furthermore, models and explanation methods should be tested on multiple datasets that represent different domains and demographic characteristics, as results derived from a single dataset may not generalize across other use cases. Testing on various datasets ensures broader applicability and fairness.

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

## A    Appendix

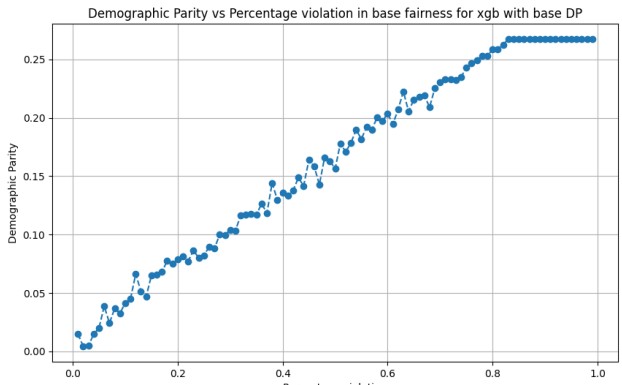

(a) Demographic parity difference for XGB model with ACSIncome

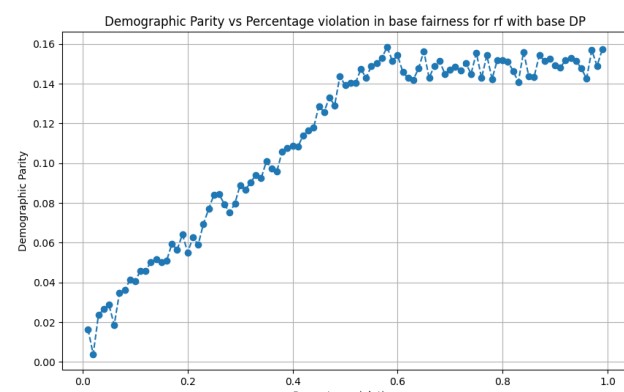

(b) Demographic parity difference for RF model with ACSEmployment

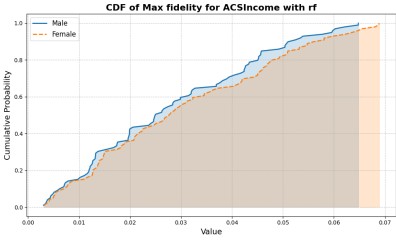
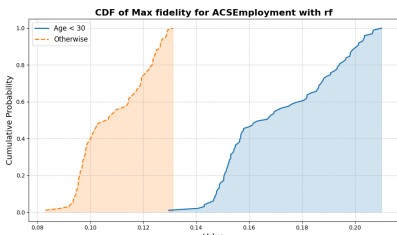
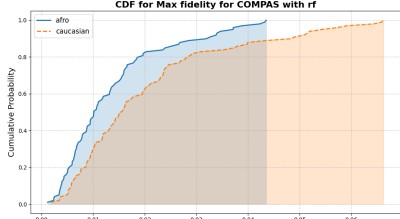

Figure 8: **CDF $\Delta Q$ with RF**

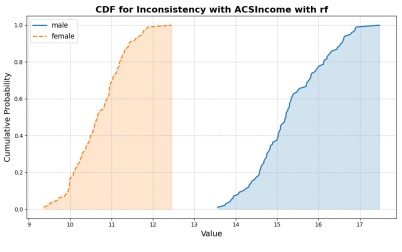
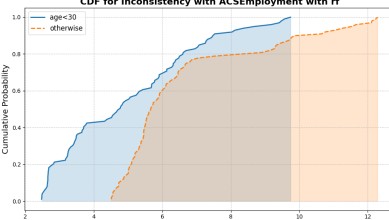
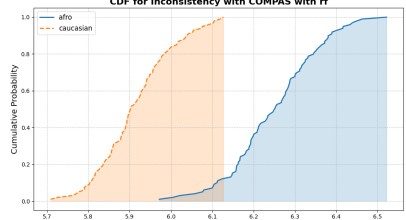

Figure 9: **CDF Inconsistency score with RF**

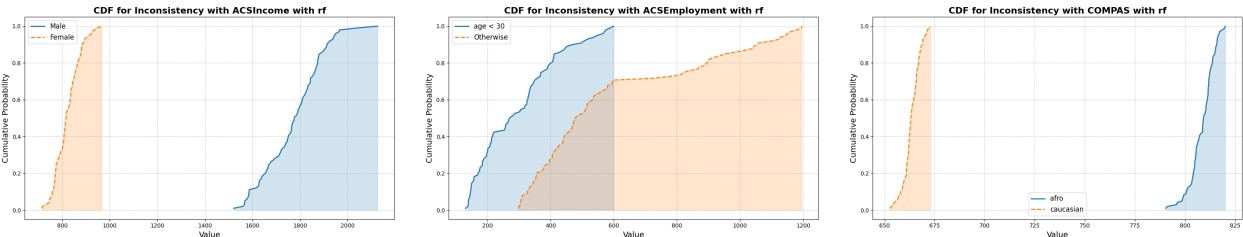

Figure 10: **CDF Instability score with RF**

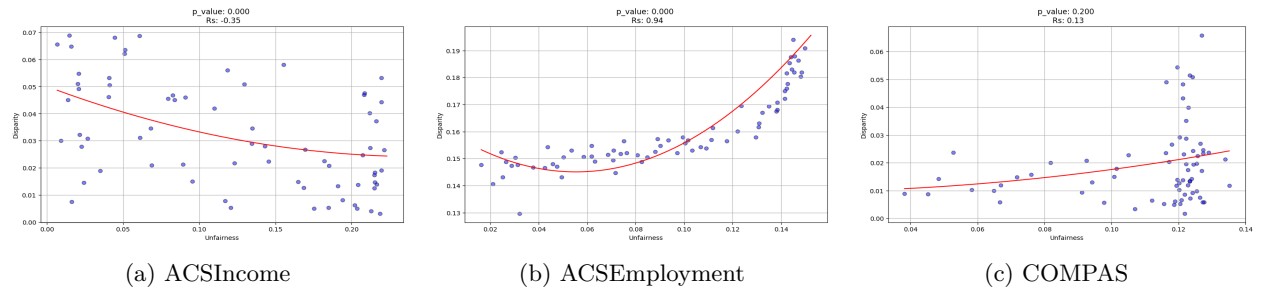

(a) ACSIncome

(b) ACSEmployment

(c) COMPAS

Figure 11: **Correlation for $\Delta Q$ and unfairness with RF**

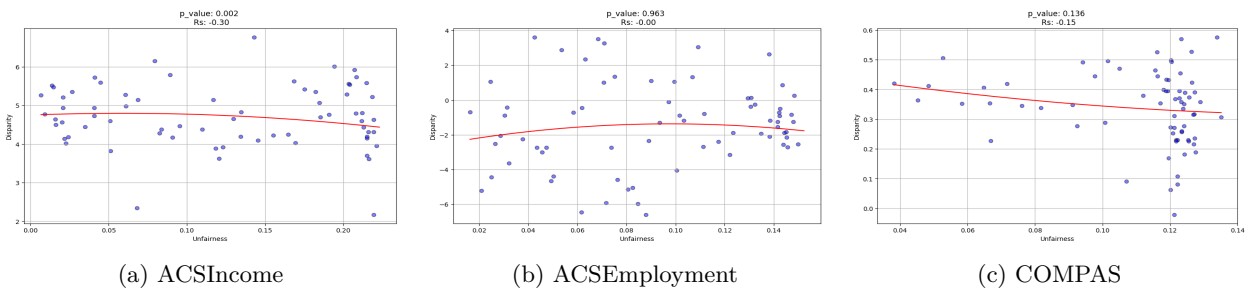

(a) ACSIncome

(b) ACSEmployment

(c) COMPAS

Figure 12: **Correlation for Inconsistency and unfairness with RF**

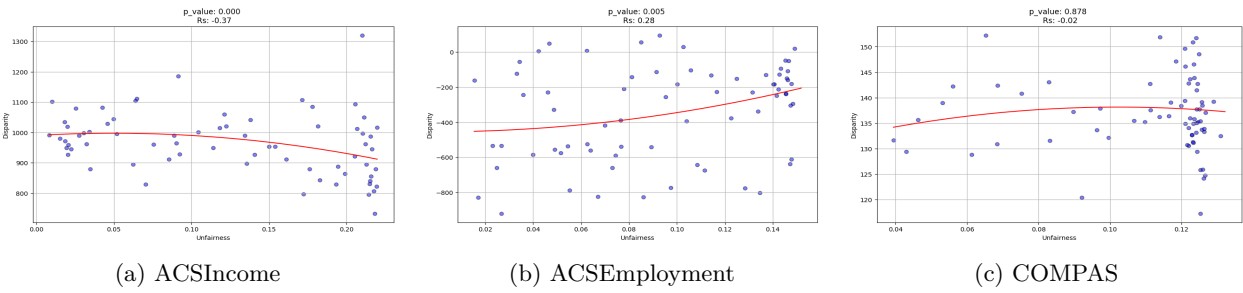

(a) ACSIncome

(b) ACSEmployment

(c) COMPAS

Figure 13: **Correlation for Instability scores and unfairness with RF**

