# OpenReview forum: "Evaluating Disparities in the Quality of Post hoc Explanations when the Explained Blackboxes are subjected to Fairness Contraints"
_TMLR — Rejected by TMLR_

### Review · Reviewer_wdA1 · 2025-08-12

**Summary Of Contributions:**

This paper investigates how fairness constraints on black-box models (XGBoost, RF) affect disparities in the quality of post-hoc explanations (LIME, KernelSHAP) across demographic groups. Three datasets are used, each with a different sensitive attribute. The authors vary fairness constraints over 99 levels and measure explanation disparities using three metrics as follows: MFGA, relative consistency, and stability. They use Mann–Whitney U tests to detect group differences and Spearman correlations to assess the relationship between unfairness and disparity. Results show dataset-dependent and model-dependent disparities, sometimes with non-linear fairness–disparity relationships, and different behavior under DPD vs. EOD constraints.

The paper addresses a highly relevant topic at the intersection of fairness and explainability, which is of particular importance in high-stakes applications. A notable strength is the granular exploration of fairness constraints through 99 different levels, offering a more detailed analysis than the common binary fair/unfair comparisons. The study uses multiple datasets and model types, which, while limited in scope, shows some attempt at generality beyond a single benchmark. The metrics employed are clearly formalized, even if some definitions warrant refinement. The results section is well organized, with tables and figures systematically arranged by metric and dataset, aiding readability and comprehension.

**Audience:**

No

**Audience Explanation:**

The intersection of fairness and explainability is an active research area, and TMLR’s audience includes many researchers interested in understanding how fairness constraints might impact explanation quality. However, while the topic is relevant, the current methodological and evidential gaps limit the reliability and impact of the findings (mentioned above), which reduces their value to the community in their present form. In addition, there is room for further improvement.

1. RQs and novelty appear only in Section 2.3 rather than in the Introduction. The Introduction reads like a literature dump with no clear logical consideration. Overcrowding of concepts without hierarchy and abrupt topic shifts break narrative flow. Inconsistent granularity is found, e.g., some methods are over-explained, others mentioned barely. Additionally, The current Section 3 hierarchy lists MFGA under 3.2, while Consistency and Stability appear as 3.3 and 3.3.1, suggesting a nested relationship that does not reflect their equal conceptual importance.

2. The second paragraph on p.2 (“It is worth noting…”) is verbose and could be expressed more concisely, while the third paragraph appears to make an argumentative claim that is not supported by subsequent discussion or evidence.

3. In Section 3.1, some bold symbols such as **n** and **a** appear only once and are never reused, which disrupts notation consistency and risks confusing readers about their significance.

**Broader Impact Concerns:**

The work is primarily an empirical analysis of fairness and explanation metrics, and thus no significant broader impact concerns arise beyond standard methodological research.

**Claims And Evidence:**

No

**Claims Explanation:**

While some results are presented clearly, key claims are not convincingly supported.

1. The perturbation scheme, e.g., replacing continuous features with the median and Gaussian noise, and categorical features with the globally least frequent category, can create group dependent out of distribution artifacts that bias disparity measurements. In datasets where feature distributions differ across demographic groups, the global median or the least frequent category may lie much farther from the typical values of one group than of another, which produces perturbed samples that are disproportionately unrealistic for certain groups. Because both LIME and KernelSHAP rely on perturbations that approximate the local data manifold, such group asymmetric distortions can degrade surrogate model fidelity more severely for one group, yielding apparent explanation disparities even when the underlying model treats groups equally.

2. Fairness constraints inevitably modify the model’s decision boundary, which can alter key performance metrics such as accuracy, AUC, and calibration. These performance shifts can in turn affect the outputs of explanation methods like LIME and KernelSHAP, since changes in predictions or local neighborhoods can influence attribution fidelity and such. If correlations between fairness level and explanation disparity are observed without reporting or controlling for these performance metrics, it is impossible to determine whether fairness itself drives the disparities or whether the effect is merely a byproduct of altered predictive performance.

3. The analysis presents no uncertainty intervals or effect sizes alongside figures or tables, making it difficult to assess the robustness or practical significance of the reported disparities.

**Requested Changes:**

Please consider the following questions to improve the current draft, taking into account the comments above.

1. Given that your perturbation scheme replaces continuous features with the global median plus Gaussian noise and categorical features with the globally least frequent category, how do you rule out the possibility that group-dependent OOD samples are driving the observed disparities? Have you conducted any experiments using perturbations that preserve group-conditional feature distributions to verify that the disparities persist?

2. What are the accuracy, AUC, and calibration results at each fairness level? Can you show that fairness–disparity correlations remain after controlling for these performance metrics? How do you justify attributing the observed disparities to fairness constraints without reporting predictive performance results, and how can you rule out that these disparities are not simply due to changes in predictive performance?

3. Why are uncertainty intervals or effect sizes not reported alongside disparity values? Without such measures, how can readers assess the robustness or practical significance of your findings?

4. Can you demonstrate that the observed fairness–disparity relationships hold for other types of explainers (e.g., gradient-based, counterfactual) and for models beyond XGBoost and Random Forests? If not, why should the conclusions be interpreted as generally applicable across explanation methods and architectures?

---

### Review · Reviewer_Hptm · 2025-08-13

**Summary Of Contributions:**

This paper investigates how fairness constraints on machine learning models influence the quality of post-hoc explanations across demographic groups. The authors focus on XGBoost and Random Forest, and employ LIME and KernelSHAP to evaluate three explanation quality metrics: Maximum Fidelity Gap from Average (MFGA), Relative Consistency, and Stability. Using three datasets (ACSIncome, ACSEmployment, and COMPAS), they examine explanation qualities and fairness achieved by Exponentiated Gradient reduction. The study finds that fairness constraints do not uniformly improve explanation equity; disparities vary by dataset, model complexity, and fairness metric. Results suggest that context-specific strategies are needed to ensure fair explanations in high-stakes applications.

**Audience:**

No

**Audience Explanation:**

Although the topic is meaningful, the findings of this paper may be of limited interest to TMLR’s audience for several reasons. First, although the paper presents quite several results and finds about the relationship between model fairness and explanation fairness, these results are fragmented and do not lead to any clear, actionable guidance. For instance, while the authors emphasize the need for context-aware metric selection to balance model performance and explanation equity, they offer no direction on how practitioners should achieve this. Second, the experimental scope is relatively narrow. The experiments are limited to two machine learning models: XGBoost and Random Forest, both of which are decision tree-based ensemble methods and share similar structural properties. This raises concerns about whether the findings can generalize to fundamentally different model architectures, particularly neural networks and deep neural networks, which dominate many modern machine learning applications and often pose more significant fairness and explainability challenges due to their high opacity. The exclusion of neural models limits the relevance of the results for a broader audience. In addition, the experiments are limited to tabular datasets and ignore more complex domains like vision or language.

**Broader Impact Concerns:**

No obvious broader impact concerns.

**Claims And Evidence:**

Yes

**Claims Explanation:**

The paper investigates the relationship between fairness constraints applied to machine learning models and fairness in post-hoc explanations. It finds that this relationship is often complex and non-linear, influenced by factors such as the dataset, the underlying machine learning model, and the explanation method used. These claims are supported by extensive empirical evaluation using three real-world datasets (ACSIncome, ACSEmployment, and COMPAS), different demographic attributes (gender, age, and race), two types of black-box models (Random Forest and XGBoost), and two widely-used perturbation-based explanation methods (LIME and KernelSHAP). Statistical robustness is ensured through hypothesis testing (Mann-Whitney U and Spearman) and reporting of p-values.

**Requested Changes:**

The introduction section lacks several essential components and should be revised. Specifically, it should clearly state the research questions, briefly outline the methodology, and summarize the main findings. Currently, some of this information appears in the Related Work section instead, which can confuse readers and obscure the paper's core contributions.

In discussing explanation methods within the introduction, it is reasonable for the authors to focus their evaluation on perturbation-based techniques, given their relevance to tabular data. However, the criticism of gradient-based and rule-based methods is not well-founded. These methods have distinct advantages in other domains, such as image and text data for gradient-based techniques, or knowledge-driven scenarios for rule-based ones.

Within the Related Work section, the statement that "this research takes a more granular approach by investigating 99 different levels of unfairness" is unclear. The paper does not define what is a "level" of unfairness until much later in the experimental section. Furthermore, the benefit of this increased granularity is not explicitly discussed or justified.

There is a redundancy between RQ1 and RQ2, as both ask how model unfairness relates to explanation quality disparities. The authors should either consolidate these questions or more clearly distinguish their scopes to avoid overlap.

In Section 3.2, the purpose of constructing the modified dataset is ambiguous. It is not clearly explained how this modified dataset is used to compute the listed evaluation metrics: Maximum Fidelity Gap from Average (MFGA), Relative Consistency, and Stability. Currently, it appears that the modified dataset is not directly used in computing these metrics, which creates confusion about its role in the methodology.

On Page 13, the statement that "XGBoost models were more sensitive to fairness metric choice than Random Forest models, likely due to their non-linear structure amplifying disparities" is misleading. Both XGBoost and Random Forest are non-linear ensemble methods.

The subsequent statement that "more complex models like XGB tend to show more pronounced non-linear relationships" is also confusing. If the authors intend to argue that XGBoost is inherently more complex than Random Forest, it should be supported with evidence or references.

---

### Review · Reviewer_BEHf · 2025-08-18

**Summary Of Contributions:**

This paper investigates an important issue at the intersection of explainable AI (XAI) and algorithmic fairness. Specifically, it explores how imposing fairness constraints on black-box models affects the fairness and quality of post-hoc explanations (LIME, KernelSHAP) for different demographic groups. The core contribution of the paper is the systematic analysis of the relationship between model unfairness and explanation quality disparities via multiple metrics across datasets. The work is well-motivated and addresses an interesting research problem. While the experimental setup is promising, the manuscript could be significantly strengthened by addressing several points regarding methodological choices, reporting of results, and overall presentation.


Key weaknesses:

**Justification and clarity of Methodological Choices**
- Bias Mitigation Technique: The paper employs a reduction-based approach for bias mitigation, as proposed by Agarwal et al. (2018). The manuscript would benefit from a clearer justification for this choice over other common techniques (e.g., pre-processing or post-processing methods). It is crucial to discuss whether the choice of this specific method could influence the observed relationship between fairness and explanation quality. A brief discussion on the potential limitations of this choice or how results might differ with other mitigation strategies would strengthen the paper's claims.

- "Levels of Fairness" Granularity: The authors create 99 "fairer" models by varying the the corresponding variable from 0.01 to 1.0 with a step size of 0.01. The rationale behind this fine-grained approach needs more explanation. Is there a practically or theoretically meaningful difference in fairness constraints between adjacent steps (e.g., 0.01 vs. 0.02)? A sensitivity analysis or justification for why this level of granularity is necessary would help readers understand the experimental design's significance. Further, it is not clear in Section 5, how the granularity of levels was leveraged for more comprehensive analysis, clear communication of this connection would be helpful to better understand the importance of the levels.

**Incompleteness in results &analysis**
Details provided in the next answer

**Audience:**

Yes

**Audience Explanation:**

XAI is a active area of research and application with ML-based tools being designed and developed for human-AI collaboration in the real world. ML tools are known to show biases and explanations are a way of bridging the gap between the human and the ML model and help the human make the best decision despite of ML shortcomings, so this is an important area of research and would be a useful source of insights for people working in this field.

**Claims And Evidence:**

No

**Claims Explanation:**

The results section currently lacks several key details required for a full interpretation of the findings, and the implications of the work. I point several out here:

- Statistical Test Reporting: For the Mann-Whitney U tests (Tables 2-4), the paper reports only p-values. To properly assess the magnitude of the observed effects, it is essential to also report an effect size metric (such as the U statistic). Please add an effect size column to these tables as it is crucial to understand and interpret the results.
- Correlation Analysis (Table 8): The derivation of this table and its takeaways are unclear. Please provide a detailed description of the analysis protocol. Specifically, please clarify:
    - What thresholds were used to classify correlations as "slight”, or "perfect"?
     - How were positive versus negative correlations interpreted in the context of the research questions? This needs to be explicitly stated and discussed.
    - Without these details, the conclusions drawn from the correlation analysis are difficult to understand.
- RQ1 Statistical Results: In the discussion of RQ1, the paper references p-values alongside Figures 1-3. However, it is not clear how these p-values were derived from the data presented in the plots. The analysis section should be expanded to explicitly describe the statistical tests performed, the hypotheses being tested, and how they relate directly to the visual evidence in the figures.

**Comments on figures**
- The current figures are lacking information required to read them and the tags and captions are quite informal. The x-axis labels are unclear in their meaning in Figures 1-3, the legend is small and the use of abbreviations (“afro”) should be avoided or clearly defined in the caption.
- Each figure caption should be self-contained and descriptive. It should explain the plot's purpose, define all its elements (e.g., explain what the "red curve" represents and how it was generated), and describe the analysis method shown briefly.

**Requested Changes:**

Based on the previous two answers, the following are critical to securing my recommendation for acceptance:
- Provide clear explanation of methodological choices (bias mitigation technique and level of fairness granularity) and discuss impact of this choice of research findings in the discussion section
- Provide complete details of the results and related analysis in Section 5 and improve the figures to clearly communicate results.,

In addition to addressing the above, I have the following comments on writing which would strengthen the work in my view

**Writing and Structural Improvements**
- Narrative Flow (Introduction vs. Related Work): There is significant thematic overlap between the last paragraph of the Introduction and Section 2.3 ("Group-Based disparities Post hoc explanation quality"). This redundancy can confuse the reader. It is recommended to move all literature review and discussion of prior work to the Related Works section. The Introduction should focus on motivating the problem, stating the paper's unique contributions, and clearly outlining the research questions.
- Placement of Research Questions: The RQs are currently located at the end of the Related Works section. Conventionally, research questions are presented at the end of the Introduction to frame the study's objectives for the reader from the outset.


**Minor Comments & Suggestions**
- Missing citation in this claim: “Moreover, recent studies have indicated that training models robust to group differences can enhance fairness by improving accuracy for the least advantaged group”
- On page 3: please introduce what “level of fairness” means
- On page 3: the term “complex black-box models” is used. Please specify what type pf complexity is being talked about and provide some examples, as there is a large variety of complex black box models in play in ML.
- Please clarify in Null hypothesis in Section 5.1.5 what is the measurement metric for model unfairness being considered. Same for 5.2
- Typo in name of Section 6.1
- Formatting of Section 5.3 is currently making it hard to parse, please clearly delineate subsection titles and the text

---

### Decision · Action_Editor_TXeq · 2025-09-17

**Recommendation:** Reject

**Additional Comments:**

The reviewers raised several methodological issues with the experiments in the manuscript. There was no response from the authors.

**Audience:**

Yes

**Audience Explanation:**

The question raised in this manuscript is of interest for the fairness and explainable AI community.

**Claims And Evidence:**

No

**Claims Explanation:**

There are sevearl issues with the experimental design that the authors have not addressed. In particular, their imputation technique can cause effects that are unrelated to the measure of interest. In addition, the authors have not provided sufficient justification for the statistical tests that they use.

**Resubmission Of Major Revision:**

The authors may consider submitting a major revision at a later time.